# A Study on the Localization of Urban Residents’ Recreation: A Moderated Mediation Model Based on Temporal Self-Regulation Theory

**DOI:** 10.3390/ijerph20065160

**Published:** 2023-03-15

**Authors:** Hui Tao, Qing Zhou, Qian Yang

**Affiliations:** 1School of Management, Minzu University of China, Beijing 100081, China; 2School of Tourism and Urban-Rural Planning, Zhejiang Gongshang University, Hangzhou 310018, China; 3School of European Studies, Beijing International Studies University, Beijing 100024, China

**Keywords:** localization of recreation, temporal self-regulation theory, sense of place, recreation involvement, recreation benefits

## Abstract

The pandemic has resulted in a further reduction in travel distance, recreational radius of destinations and other levels of tourism activity, making “local people traveling locally” a new feature. From the perspective of localization of urban residents’ recreation, this paper describes a moderated mediation model based on temporal self-regulation theory. Five representative urban parks in Beijing were selected as study areas, and data collected through a questionnaire were used to discuss the behavioral characteristics of localized recreation and the formation mechanism of sense of place among urban residents in Beijing. The results showed that: (1) connectedness beliefs and temporal valuations positively influenced sense of place, and had a positive indirect effect on sense of place through the mediating role of recreation involvement; (2) recreation benefits positively influenced sense of place; (3) recreation benefits reinforced the direct and mediating role of recreation involvement. Based on these findings, the paper concludes with a discussion of the theoretical value and practical implications, as well as future research directions for park and city management.

## 1. Introduction

The fast-paced lifestyle, intense work pressure and recurrent outbreaks of COVID-19 have increased the overall negative emotions and psychological needs of individuals. Meanwhile, the demand of residents for local travel and daily leisure is greatly increasing due to the restrictions on cross-regional movement of people caused by the epidemic prevention and control measures. According to the *Annual Report of China Domestic Tourism Development (2022–2023)*, the pandemic has made residents more cautious in their travels, which is reflected in the obvious reduction in travel distance and recreational radius of destinations, showing new characteristics such as short time, close distance and high frequency. The urban recreation market will enter the new stage of development in which the service targets are mainly local residents.

“Recreation” consists of something carried out for refreshment or diversion, it is an activity that renews one’s health and spirits by relaxation and amusement [1]. The localization of recreation emphasizes recreational activities performed “in situ” (workplace, living place or recent permanent residence) from the geospatial dimension. Unlike long-distance travel on holidays, daily recreational needs can be met in the inner space or suburbs of cities. Meanwhile, the localization attaches great importance to placeness, which is regarded as a quality that distinguishes a place from other places and as a significant consumption element for recreationists. As places become more homogeneous and standardized, their distinctiveness and diversity are weakened, making it difficult for recreationists to form local identity and emotional attachment [2]. In the context of localization of recreation, through repeated interaction and complex connections between people and places, individuals are placed in familiar and meaningful environments, with local memories and emotions, resulting in the transformation of places from physical space to humanized emotional space. From this perspective, the sense of place is formed under the influence of placeness and its constructability. On the one hand, it can create a unique local image and emotional quality to satisfy the physiological and psychological needs of recreationists; on the other hand, it can enhance the sense of belonging and responsibility of recreationists, and stimulate their willingness to revisit and recommend the place. In addition, the localization of recreation is not just the exclusive discourse of local residents. In recent years, tourism has been showing the characteristics of “localization”, in which tourists show a tendency to converge with local residents in terms of preferences and behavioral patterns. An increasing number of tourists are widely integrated into the public space of destinations to experience the recreational activities preferred by local residents.

Temporal self-regulation theory (TST) has been described as a “viable, integrative framework for contemporary research” that synthesizes ideas from cognitive psychology, behavioral economics and neuroscience into a relatively comprehensive “bio-psycho-social” model, which explains the multiple factors that influence people’s health behaviors [3,4]. According to TST, the capacity to be involved in behavior in accordance with long-range interests arises from a complex combination of biological, cognitive and social factors. With respect to personal behavioral choices, individuals intend to pursue behaviors that they believe are likely to have positive, immediate consequences. Individual differences in time perspective are associated with health-relevant decision-making processes [5,6]. TST has been widely used to explain and predict the occurrence of and change in individual health behaviors, including episodic drinking [7], sugar-sweetened beverage consumption [8], everyday smoking [9], supplement use [10] and medication adherence [11]. There is a strong link between health behaviors and personal choices of recreational activities [12]. Empirical studies in environmental and public health have shown that frequent participation in recreational activities can effectively reduce the risk of mental disorders and chronic diseases.

Recreational behavior is a typical personal decision-making behavior in a healthy life. TST allows for the good explanation and prediction of health behaviors. Similarities in underlying mechanisms of health and recreational behavior, as well as evidence in both fields for the importance of TST, indicate that the theory can be applied to recreational behavior. Hence, applying the theory to assess recreational behavior may allow for simultaneous assessment of structures that have an impact on behavior and provide a comprehensive explanation of the issue. Residents’ sense of place is a complicated and dynamic process that results from the long-term interaction between residents and recreation areas [13]. This study constructed a moderated mediation model based on TST from the perspective of localization of recreation to investigate the behavioral characteristics of localized recreation and the formation mechanism of sense of place among urban residents. Using TST to explore the sense of place may be an essential process for identifying the potential relationships between important variables. The contributions of this study are as follows: first, in theory, it develops and extends the application of TST in the field of recreation behavior by applying it to the study of localization of recreation among urban residents; at the same time, it explores in depth the relationship among recreation involvement, recreation benefits and sense of place, which provides a powerful complement to urban park research from the perspective of demand. Second, in practice, this study explores important factors in the behavioral intention and decision-making process of localized recreation of residents, which helps to provide theoretical support and management suggestions for urban planning departments to reasonably meet the needs of localized recreation and improve the construction of localized recreation.

## 2. Literature Review

### 2.1. Sense of Place

The concept of sense of place reflects the emotional, symbolic and spiritual aspects of places. It implies a relational notion of place, according to which undifferentiated space becomes place when we endow it with value [14]. In human geography, the term is used to encompass all the subjective meanings that become attached in some way to a place, identification with a place therefore grows through psychological investment and repeated encounters over time, leading to the gradual accumulation of meanings for that place, which in turn contributes to a sense of self and belonging [15]. As a complex affective bond between people and a specific location, sense of place should be seen as socially constructed, relational and part of social interactions and wider social processes [16], for example, a place may have restorative properties that enable individuals to get away from everyday routines or provide spiritual fulfillment.

Some scholars have explored the variables that influence recreationists’ sense of place, such as: recreation involvement, recreation benefits. Involvement is regarded as a motivational variable reflecting the extent of personal relevance of the decision to the individual in terms of basic goals, values and self-concept. Manfredo defined involvement as the degree of interest in an activity and the affective response associated with that interest [17]. The construct of involvement in this study is conceptualized to be reflective of an enduring trait. It can reflect the level of commitment to recreation activities, related products or experience, and is regarded as an important concept for understanding recreation behavior [18]. When an individual has pleasant experiences from recreation participation and regards it as an important form of recreational life, recreation involvement arises as a state of motivation, incentive or interest, which persists in the whole process of activity participation and always brings benefits to the recreation subject. Recreation benefits are the main motivation for individuals to go out for recreation, as individuals gain continuous satisfaction by participating in any enjoyable recreational activities, being a subjective factor that helps to improve physical and mental conditions or satisfy needs by means of recreation behavior. Participation in recreational activities can alleviate depression, reduce stress, boost happiness, improve interpersonal relationships and enhance self-efficacy, all of which stimulate intrinsic desire of individuals to improve life satisfaction, and people will pursue the benefits of recreation at both individual and society levels [19]. After achieving the objective of recreation, the subjective feeling for recreation benefits is obtained, and the greater the benefits received, the more active the participation behavior will be [20]. To sum up, this study attempts to further investigate the formation mechanism of the sense of place based on recreation involvement and recreation benefits from the perspective of localization of recreation.

### 2.2. Temporal Self-Regulation Theory

According to Bandura’s social cognitive theory (SCT), self-regulation is the interaction of the individual, behavior and environment, and it refers to the process of self-generated thoughts, feelings and actions that are planned and cyclically adapted to the attainment of personal goals [21,22]. Self-regulation is cyclical, as personal, behavioral and environmental factors are constantly changing during the course of learning and performance, and the feedback from prior performance is used to make adjustments during current efforts [23]. However, most studies have found that health-related behaviors are not completely under intentional control [24]. There has been a growing awareness of the influence of temporal factors on the decision-making process itself. Individuals’ behavioral choices are limited by temporal factors, the rationality of behavior largely depends on the temporal frame adopted, and long-term factors can motivate people to engage in health behaviors [25]. Therefore, the TST, which is based on the “intention–behavior” link and emphasizes the combined effects of temporality, behavioral prepotency and self-regulatory capacity, appears to be suitable for exploring the localization of urban residents’ recreation and the formation mechanism of sense of place.

In the motivational stage, intentions to engage in a certain behavior are driven by connectedness beliefs and temporal valuations. Within the motivational sphere, connectedness beliefs arise from the strength of connectedness between the behavior itself and the outcomes, and the stronger the connectedness between the two, the more likely the subject is to develop connectedness beliefs. Temporal valuations inform intention to perform or refrain from behavior [26]. For example, most health protective behaviors such as exercise and healthy eating are beneficial in the long term if performed consistently.

Evidence increasingly suggests that self-regulation, defined as the capacity to manage and control cognitive, behavioral and emotional responses to internal or environmental cues, is also an important component of health behavior theory [27]. Since the costs and benefits of health protective behaviors and health risk behaviors differ in temporal dispersion, adequate self-regulatory capacity is thought to be quite relevant to health behaviors. Adherence to healthy lifestyle behaviors requires planning and the ability to adapt to changes in the environment, these are all contained within the concept of self-regulation. Behavioral prepotency refers to the likelihood of performing a behavior given the frequency of performance in the past, habit strength and salient environmental cues; habit strength is one component of behavioral prepotency and is considered a strong predictor of behavior [28]. Specifically, executive function tests such as the Go/No-Go test are used. In addition, the TST purports that the influences on the intention–behavior relationship differ depending on the environmental context present at the time of performance. If the environment were perceived as being supportive of behavioral performance, then behavioral performance would be less reliant upon intentions and self-regulation than if the same behavior were performed in an environment that was highly distracting and unsupportive. However, for individuals who experience the immediate environment as unsupportive for behavioral performance, their behavior is determined by intentions, behavioral prepotency and self-regulation [29] (Figure 1).

## 3. Research Hypotheses

By specifying behavioral self-regulation as influenced by both motivational and momentary factors, TST not only offers the potential to improve the prediction of behavior but also to identify pre- and postintentional factors influencing successful behavioral self-regulation which could be targeted by interventions. In light of previous research and the tenets of TST, this study aims to explore the constructs of TST as predictors of sense of place. Five variables are integrated into the TST framework, namely, recreation involvement into intention valuation, recreation benefits into perception evaluations where behavioral prepotency and self-regulatory capacity work together and sense of place into behavioral intention.

### 3.1. Connectedness Beliefs, Temporal Valuations and Sense of Place

The motivational sphere describes conscious deliberations whether or not to engage in a behavior, this includes, in particular, temporal valuations about the time points when positive or negative consequences of the behavior can be expected. These temporal valuations inform intention to perform or refrain from behavior. The sense of place reflects a deep emotional connection between individuals and places. The perception and impression of recreation subjects for the space are a kind of local meaning that people give to the space, which endows places with recreational characteristics that are different from other spaces. Hammitt found that the more frequently individuals use a place, the stronger their sense of place, as well as their level of identification with the place [30]. Manzo pointed out that the sense of place has a significant incentive for individuals to make greater use of public places and spend more time on social interactions [31]. According to TST, connectedness beliefs and temporal valuations are at play in the motivational spectrum, behavior takes on a kind of psychological inertia as performance is repeated over time and habit strength is generated when behaviors are performed with high frequency in stable situational contexts [32]. Min Xiangxiao found that the mechanism of action between tourism image and sense of place was similar to the mechanism between “cognition and emotion” in psychology. Perception lies at the point of action between the two. When receiving external stimuli, recreationists make value judgments through the process of perception, thus ascending to a stable and abstract emotional experience and forming a sense of place [33]. On this basis, the following hypotheses are proposed in this study:

**H1a.** 
*Connectedness beliefs positively influence sense of place.*


**H1b.** 
*Temporal valuations positively influence sense of place.*


### 3.2. Connectedness Beliefs, Temporal Valuations and Recreation Involvement

Involvement is essentially an attitude toward recreation that represents a determination about what is important, meaningful or relevant and can be used to explain participants’ recreation decisions and the process of decision-making [34]. Havitz and Dimanche defined recreation involvement as an unobservable state of motivation, arousal or interest toward a recreational activity, place and facility. The motivational sphere describes conscious deliberations whether or not to engage in a behavior and connectedness beliefs and temporal valuations in TST models form the main determinants of intention. This sphere of influence is similar to SCT in that it results in a deliberate decision to become involved in behavior or intention [35]. According to SCT and TST, those with high-outcome expectancy are more motivated to engage in the behavior, and there is indeed a causal relationship between outcome expectancy and formation of intention to engage in health behaviors. In addition, several studies have shown that increasing the salience of the association between current behavior and later outcomes has a strong motivational effect on intentional behavior [36]. Therefore, the following hypotheses are proposed:

**H2a.** 
*Connectedness beliefs positively influence recreation involvement.*


**H2b.** 
*Temporal valuations positively influence recreation involvement.*


### 3.3. Recreation Involvement and Sense of Place

In the TST, frequency of past behavior can be a proxy for behavioral prepotency, and past behavior consistently appears to be the best predictor of future behavior [37]. The relationship between involvement and sense of place has been extensively researched by scholars, who have found a significant positive correlation between recreation involvement and sense of place. Recreation involvement predicts the level of sense of place and influences people’s emotional attachment to a place of recreation. Sense of place also reflects the level of involvement in recreation and influences people’s related behavior in recreation areas [38]. Havitz et al. proposed that recreation involvement had a driving effect on sense of place, emphasizing individuals’ sense of participation and experience during recreation, and their hope to enrich their spiritual world through recreation, which reflects emotional factors of recreationists about places from the perspective of involvement and can deeply explore the influence mechanisms of recreationists’ sense of place [39]. Wang Kun et al. proposed that involvement has a direct impact on sense of place [40], and Chen Wanru et al. transferred the model “leisure involvement → sense of place” to urban space for daily recreation activities [41]. Hence, the following hypothesis is formulated:

**H3.** 
*Recreation involvement positively influences sense of place.*


### 3.4. The Mediating Role of Recreation Involvement

The relative predictability of behavior from intention, self-regulatory abilities and behavioral prepotency will depend on the ambient contingency structure of the social and physical environment in which the behavior occurs, in a manner consistent with the conceptualization of Rothman of the change process, all of these relationships are conceptualized within TST to represent cumulative feedback loops [42]. Involvement theory shows that recreation involvement is a mediating variable between intentions and behavioral intentions. The higher the degree of individual involvement in recreation activities, the stronger the “intention–behavior” link, and the more salient the participation behavior, recreation intentions and behavioral intentions of the subject [43]. Guo Qigui et al. examined the role of involvement as a mediating variable in the influence of recreation motivation on satisfaction [44]; Wang Zhenning et al. examined the correlation among motivation, involvement and behavioral intention of urban residents in self-driving travel, and the results showed that involvement played a partial mediating role in the influence of motivation on behavioral intention [45]; Fan Mengdan found that rural imagery had a significant positive influence on the sense of place through the mediating role of involvement, taking B&Bs in Xiamen as an example. Therefore, the following hypotheses are proposed:

**H4a.** 
*Connectedness beliefs positively influence sense of place through the mediating role of recreation involvement.*


**H4b.** 
*Temporal valuations positively influence sense of place through the mediating role of recreation involvement.*


### 3.5. Recreation Benefits and Sense of Place

Some scholars believe that the measurement of recreation benefits should be integrated into a systematic evaluation model. Recreation has significant benefits for individual physical health (e.g., improving physical fitness, relieving stress, etc.), while implicit benefits are invisible satisfaction and growth in psychological states, and participants are more active in recreation activities when the benefits are highly evaluated [46]. Recreation benefits, as the subjective evaluation of recreationists’ perceptions about their experience during recreation activities, are closely related to the sense of place. Bi Lu Luan et al. argued that participating in recreational sport had the benefit of motivating individuals to interact with places, and that recreation benefits of different experiences were the driving force of intention to revisit [47]. Recreational activities and experience affect recreationists’ perceptions of local meaning, and to some extent, affect the level of well-being of recreationists in many ways, both physically and mentally, and in social relationships [48]. Thus, the following hypothesis is proposed:

**H5.** 
*Recreation benefits positively influence sense of place.*


### 3.6. The Moderating Role of Recreation Benefits

Individual differences in responsiveness to environmental triggers should determine which variables—self-regulation or behavioral prepotency—are more likely to predict an ability to behave in a manner that is consistent with intentions to maintain a healthy lifestyle [49,50]. From a subjective and empirical perspective, recreation benefits are a strong predictor of participation in recreation activities [51]. Recreation benefits represent the achievement of recreation goals, and participants’ psychological needs are met and they feel the benefits of recreation after participating in recreation activities [52]. Tinsley et al. argued from an individual perspective that the higher the participants’ evaluation of potential benefits from recreation activities, the more obvious the attitude of participation and the more active the behavioral performance of participation [53]; similarly, studies have been conducted in Chongqing, a popular city in China, demonstrating the moderating role of recreation benefits in the relationship between behavior and sense of place [54]. Referring to the role of behavioral prepotency and self-regulatory capacity in the TST framework, this study uses recreation benefits as a moderating variable and proposes the following hypotheses:

**H6.** 
*Recreation benefits play a moderating role in the relationship between recreation involvement and sense of place, i.e., the more recreation benefits recreationists receive, the stronger the positive relationship between recreation involvement and sense of place, and vice versa.*


### 3.7. A Moderated Mediation Role

Those who experience positive outcomes may come to believe that the balance of future costs and benefits of continuation of the behavioral change (i.e., maintenance) will be worth it; that is, the likelihood of positive outcomes given the next performance of the behavior is strengthened, or value of the future experience of the outcome is enhanced by prior experience. As such, TST captures the influence of experiential aspects of the behavior change process on future behavior change efforts [55]. Edwards and Lambert showed that a moderated mediation effect may be established when the theoretical mechanisms of the moderating effect and the mediating effect work together [56]. From the above hypotheses, it is clear that connectedness beliefs and temporal valuations positively influence sense of place through the mediating mechanism of recreation involvement, and recreation benefits help to strengthen the relationship between recreation involvement and sense of place. In summary, recreation benefits may moderate the two mediating paths, namely, “connectedness beliefs → recreation involvement → sense of place” and “temporal valuations → recreation involvement → sense of place”, that is, a mediating effect may be moderated as well. Thus, the following hypotheses are proposed:

**H7a.** 
*Recreation benefits moderate the mediating effect of recreation involvement between connectedness beliefs and sense of place, i.e., the more recreation benefits recreationists receive, the stronger the mediating effect of recreation involvement between connectedness beliefs and sense of place; and vice versa.*


**H7b.** 
*Recreation benefits moderate the mediating effect of recreation involvement between temporal valuations and sense of place, i.e., the more recreation benefits recreationists obtain, the stronger the mediating effect of recreation involvement between temporal valuations and sense of place; and vice versa.*


To sum up, this paper has constructed a moderated mediation model based on TST, and the hypothetical model is shown in Figure 2.

## 4. Study Design

### 4.1. Scale Development

The research questionnaire is divided into two parts. The first part is the central part of the questionnaire, including the scales of different variables. The measurement items of each variable in the model are from mature scales widely used in the relevant literature. Among them are (1) connectedness beliefs scales, which refer to the research of Hsu [57] and Chiu [58], four items in total; (2) temporal valuations, which mainly use the CFC scales developed by Feng Jiaxi [59], three items in total; (3) recreation involvement variable, which mainly use the involvement scales developed by Li Qun [60], six items in total; (4) recreation benefits, mainly based on the recreation benefit scales developed by Yin Jianjun [61], five items in total; (5) sense of place variable, which refers to the research of Liu Qunyue [62], six items in total. All variables were measured using a 5-point Likert scale. The second part is the personal information of tourists, which contained three demographic characteristics and four recreation behavior characteristics. The characteristics of individual recreation behavior were used as the main control variables [63].

### 4.2. Case Parks

Due to the impact of COVID-19, recreation in parks is becoming a new way of life, the placeness contained in urban parks means important emotional significance to residents. We chose to collect data in urban parks of Beijing for two reasons: (1) Beijing has many urban parks which share obvious characteristics of large scale, diversity and balanced development, providing both tourism services for foreigners and leisure services for local residents [64]. (2) The five case parks attract crowds of recreationists and serve a wide range of people, forming high-quality places for recreation with certain social influence. Therefore, they are very representative and typical in reflecting recreation behaviors.

Beijing Municipal Forestry and Parks Bureau has classified urban parks into: comprehensive parks, community parks, historical parks, ecological parks and cultural theme parks according to the main functions undertaken by the parks, with reference to *Urban Green Space Classification Standard* (CJJ/T85-2002) and *Regulations of Beijing Municipal Parks*. Specifically, comprehensive parks refer to parks with complete functions, well-equipped facilities and rich content, which can satisfy the diverse needs of different groups of visitors. Community parks refer to parks with necessary supporting facilities and activity areas, mainly serving the residents within a certain residential area for daily leisure activities. Historical parks refer to parks with outstanding historical and cultural value, which have had an impact on the urban transformation or cultural and artistic development of the city. Ecological parks refer to parks with natural environments characterized by original ecology or low human interference, which focus on meeting visitors’ needs to get close to nature, including forest parks, suburban parks, wetland parks, etc. Cultural theme parks refer to parks with special themes or cultures as their core content, including theme parks, botanical gardens, zoos and amusement parks.

Based on the classification of urban parks in Beijing by Tao Xiaoli [65], five representative urban parks were selected as study areas, namely: comprehensive park (The Summer Palace), community park (The Black Bamboo Park), historical park (Temple of Heaven), ecological park (Olympic Forest Park) and cultural theme park (China National Botanical Garden). The locations of the case parks are shown in Figure 3.

### 4.3. Data Collection

In the process of scale design, in order to ensure the accuracy and applicability of the scale, a presurvey was conducted with a total of 100 questionnaires distributed in the case parks from 6 April 2022 to 10 April 2022. The formal questionnaire was finally designed after analyzing the presurvey, deleting or revising ambiguous and unclear items. The presurvey results show that the Cronbach’s alpha of each construct is greater than 0.7, indicating that the scale has good reliability; the standardized factor loading values of each item are above 0.6, indicating that the scale has good construct validity.

The formal survey was carried out from 13 April 2022 to 28 April 2022. Recreationists were chosen using systematic random sampling in areas with numerous recreationists, every five recreationists passing through the areas were approached. Before conducting the questionnaire research, an optional question of “Are you a local resident?” was set to filter out non-target objects. After the recreationists had accepted the invitation, one research assistant informed them of the purpose of the survey, the confidentiality of information and the meanings of some incomprehensible concepts, and all chosen recreationists agreed to participate. A total of 600 questionnaires were collected, and after eliminating invalid data, 545 valid samples were obtained, with a completion rate of 90.83%.

### 4.4. Sample Characteristics

The statistical results of demographic characteristics are presented in Table 1. It can be seen that among the valid samples, female respondents (50.5%) marginally outnumbered their male counterparts (49.5%), but the gender distribution was relatively even. The highest proportion of participants was aged 19 to 30, followed by those aged 31 to 45. Regarding physical condition, 36.5% of recreationists thought that they were in good health and 37.8% assessed their health as fair.

The descriptive statistical analysis of recreation behavior characteristics (Table 2) showed that: recreationists spent mainly 1–2 h in the park, accounting for 35.2%; most recreationists engaged in recreational activities in the park less than 3 times or 3–15 times per month, accounting for 64.6%; traveling alone was the main travel mode, accounting for 30.6%; recreationists were mainly visitors in the neighborhood, accounting for 55.6%, followed by long-distance travelers (44.4%).

## 5. Results

### 5.1. Measurement Model

The 545 valid questionnaires were examined concerning their reliability using SPSS 25.0, and the results are shown in Table 3. Cronbach’s alpha of the five dimensions was greater than 0.7, so this questionnaire indicated a relatively high level of stability and internal consistency. The confirmatory factor analysis (CFA) was used to test the model fit and verify the accuracy of this structure. χ^2^/df = 3.285 < 5, RMSEA = 0.065 < 0.08; RFI = 0.904, CFI = 0.940, NFI = 0.916, IFI = 0.940, TLI = 0.931, all of which were greater than 0.90. The fit indexes all met the fit criteria, thus the goodness of fit was high in the overall model, and the model is valid.

The standardized factor loading values (24 items in the scale) were all greater than 0.6, which meant that the loading for each item met the criteria. The composite reliability values (five variables) were all greater than the acceptable threshold of 0.7, indicating that the latent variables showed good construct reliability. The average variance extracted results for five variables were all above 0.5, demonstrating good convergent validity in the study, and that the dimensions selected for the questionnaire could well explain the variance of the variables.

It can be seen from the discriminant validity (Table 4) that there was a significant correlation (*p* < 0.001) among variables, all of which were less than the square root of AVE, indicating that the latent variables were correlated and discriminated from each other, and the discriminant validity of the scale was ideal.

### 5.2. Structural Model

To test the research hypotheses, multiple linear regression analysis was performed on the sample data, and the VIF values of all regression models were less than 5, indicating that multicollinearity was not a serious concern in the models. Meanwhile, the interaction terms involved were all centered in order to avoid potential threat of multicollinearity.
(1)Main effects. To test H1a and H1b, sense of place was first included as the dependent variable, and then the control variable and independent variable were added into the regression equation. As shown from model 2 in Table 5: connectedness beliefs had a significant positive effect on sense of place (β = 0.382, *p* < 0.001); temporal valuations had a significant positive effect on sense of place (β = 0.366, *p* < 0.001). Therefore, H1a and H1b were supported. As shown in model 3: recreation benefits had a significant positive effect on sense of place (β = 0.591, *p* < 0.001). Thus, H5 was supported.(2)Mediating effects. This study followed the steps proposed by Baron and Kenny to test the mediating effect of recreation involvement. As shown in model 8: connectedness beliefs had a significant positive effect on recreation involvement (β = 0.338, *p* < 0.001); temporal valuations had a significant positive effect on recreation involvement (β = 0.249, *p* < 0.001); therefore, H2a and H2b were supported. Model 4 was used to test the relationship between recreation involvement (the mediating variable) and sense of place (the dependent variable). Model 4 showed that recreation involvement had a significant positive effect on sense of place (β = 0.585, *p* < 0.001). Therefore, H3 was supported. Meanwhile, as shown in model 2, connectedness beliefs and temporal valuations had a significant positive effect on sense of place, and it can be seen from model 5 that after adding recreation involvement as a mediating variable, the effect of connectedness beliefs and temporal valuations on sense of place was still significant (connectedness beliefs: β = 0.252, *p* < 0.001; temporal valuations: β = 0.271, *p* < 0.001), but compared to model 2, there was a greater decrease in their values. In other words, recreation involvement played a partial mediating role in the relationship among connectedness beliefs, temporal valuations and sense of place. Therefore, H4a and H4b were supported.(3)Moderating effects. As shown in model 6, recreation involvement had a significant positive effect on sense of place (β = 0.312, *p* < 0.001); the interaction term between recreation involvement and recreation benefits had a significant positive effect on sense of place (β = 0.237, *p* < 0.001), indicating that recreation benefits positively moderate the relationship between recreation involvement and sense of place. Therefore, H6 was supported.

In order to test the moderating effect of recreation benefits, the method of Aiken and West was used to demonstrate the consistency of the moderating effect with the research hypotheses. Figure 4 displays the differences in the effect of recreation involvement on sense of place at different levels of recreation benefits, and it can be found that the slope of high-level recreation benefits was greater than that of low-level recreation benefits. Therefore, H6 was also supported.
(4)Moderated mediation effects. The regression coefficients and 95% confidence intervals were estimated using the bootstrapping method according to the test for moderated mediation effects proposed by Hayes, and the simple slope was used to test interaction effects of the moderating variables with the mediating variables, thus finally obtaining the changes after the mediating effects were moderated (a total of 5000 bootstrap samples were chosen). As shown in Table 6, when recreation benefits took a low value of −1.087, that is, when recreation benefits were within one standard deviation below the mean (M − 1SD), the value of the effect of connectedness beliefs on sense of place through recreation involvement was 0.0104, and the 95% bootstrap confidence interval was [−0.0018, 0.036], containing zero; when recreation benefits took a high value of 1.087, that is, when recreation benefits were within one standard deviation above the mean (M + 1SD), the value of the effect of connectedness beliefs on sense of place through recreation involvement is 0.0972, and the 95% bootstrap confidence interval was [0534, 0.152], not including zero. The confidence intervals ranged from including zero to not including zero, indicating that recreation benefits had a moderating effect in the mediation path “connectedness beliefs → recreation involvement → sense of place”. Thus, H7a was supported. Similarly, it can be found that recreation benefits played a moderating role in the mediation path “temporal valuations → recreation involvement → sense of place”, thus H7b was supported.

## 6. Conclusions and Discussion

### 6.1. Conclusions

In this paper, a moderated mediation model based on TST was constructed to explore the effect of connectedness beliefs and temporal valuations on sense of place, and to examine the mediating and moderating roles of recreation involvement and recreation benefits in the relationships. The study showed that:(1)Connectedness beliefs and temporal valuations positively influenced sense of place, and they also had a positive indirect effect on sense of place through the mediating variable of recreation involvement. Given that a large number of scholars have verified the significant positive correlation between recreation involvement and sense of place [66,67], this study added two antecedent variables, connectedness beliefs and temporal valuations, based on the TST framework. Recreation involvement served as a significant predictor of sense of place and also as a mediating variable among connectedness beliefs, temporal valuations and sense of place. In recreational activities, the stronger the connectedness between present actions and anticipated outcomes, and the closer the values attached to temporally dispersed outcomes, the greater the sense of involvement in the activity and the more likely it is to generate a sense of place.(2)Recreation benefits could significantly predict and positively influence sense of place. Greater recreation benefits increase the probability and degree of sense of place. Recreation benefits, as subjective evaluations of the individuals’ perceptions about the degree of satisfaction they achieve from recreational activities, can significantly influence participants’ attitudinal dispositions and behavioral performance. As the quality of recreation improves, residents feel a greater sense of dependence and identification with the recreation site, and they give positive feedback to the sense of place through feedback mechanisms, which will eventually manifest itself in residents’ attitudes towards choosing this place for ongoing recreation behavior.(3)The moderating role of recreation benefits. This study examined the moderating role of recreation benefits in the paths, which could strengthen the positive relationship between recreation involvement and sense of place, that is, the more recreation benefits recreationists received, the stronger the positive effect of recreation involvement on sense of place. In addition, recreation benefits mediate the role of recreation involvement in mediating between connectedness beliefs, temporal valuations and sense of place, and the more recreation benefits recreationists received, the stronger the mediating role of recreation involvement in the relationship among connectedness beliefs, temporal valuations and sense of place.

### 6.2. Theoretical Implications

The theoretical contributions of this study were manifested as follows:

First, a TST model was first introduced into the research of recreation behavior, and recreation involvement, recreation benefits and sense of place variables were applied to the context of localization of urban residents’ recreation, enriching the TST empirical research results. On the one hand, individuals usually believe that recreation in parks can help individuals to improve their sense of well-being, belonging and attachment, which implies connectedness beliefs about park recreation. However, such behavior must last for a relatively long time in order to obtain these values or benefits, and individuals have to suffer from problems such as long travel distance, inconvenient transportation and lack of landscapes. As the values or benefits are not close in time, temporal valuations decrease and the degree of individual recreation involvement weakens as well. Past recreation behaviors in parks have brought benefits such as promoting physical and mental health and building social interactions, and these recreation benefits can boost individual recreation involvement to positively influence the formation of sense of place. After measuring the overall benefits and costs of recreation behaviors in parks, individuals reinforce the effects of recreation behaviors by themselves and strengthen the positive influence of each variable on sense of place. On the other hand, the flow of sense of place in TST is from one context to the next. Individuals have focused on temporal valuations, connectedness beliefs and recreation involvement for the next behavior after engaging in a successful recreation activity, and have stored recreation benefits such as pleasure and self-identity for themselves in the form of symbols, thus individuals have acquired and retained a sense of place, which will influence the recurrence of recreation behaviors.

Second, urban parks were used as study areas to explore the mechanisms of generating a sense of place for residents. While studies on sense of place based on psychology explore more about people’s relationships with relatively unfamiliar environments such as tourist destinations, studies on sense of place based on phenomenology focus more on people’s relationships with relatively familiar environments such as home and neighborhood communities. However, there are also many specific “places” that are between familiar and unfamiliar to people, which also deserve careful study, and urban parks are such special places, which are currently the subject of very few studies [68]. The process of localization takes place in the context of the extensive connection between people and places, whose abundant habitual practices constitute substantial internal diversity, while the habitual practices based on places are intimately embedded in a broad spatial structure of social relations. This brings up an implicit question of the relationship between people and place: what sense of place is created in public recreational spaces such as urban parks? Based on the path of an empirical psychological approach, this study explores the mechanisms influencing the localization of recreation and the generation of sense of place among residents in five representative urban parks in Beijing, providing a powerful complement to urban park research from the perspective of demand.

Finally, using recreation benefits as a moderating variable, the mediation process between recreation involvement and sense of place was explained based on the perspective of recreation benefits. Recreation benefits, as the subjective evaluation of recreationists’ perceptions about their experience during recreation activities, are not only closely related to recreation involvement, but also have a positive impact on the sense of place. While existing studies tend to construct causal models with recreation benefits as dependent or mediating variables, this study argues that the moderating effect of recreation benefits needs to be further explored. Empirical evidence has found that recreation benefits could strengthen the positive relationship between recreation involvement and sense of place, and the more recreation benefits recreationists received, the stronger the mediating role of recreation involvement in the relationship among connectedness beliefs, temporal valuations and sense of place. This study uses recreation benefits as a moderating variable in the empirical analysis, reconstructing the connotation and extension of recreation benefits and providing new ideas for the boundary limitation of subsequent recreation benefits research.

### 6.3. Planning and Management Implications

Based on the TST framework, this study discusses the practical implications of localized recreation for park and city management.

First, as for park management, it is necessary (1) to innovate the interactive design of recreation space and set up interactive activities, such as horticulture, plant maintenance, landscape engineering and other gardening activities that recreationists can participate in. This can increase the degree of involvement and enthusiasm for interaction, and guide individuals to experience physical senses and internal emotions, thus enhancing recreation benefits for recreationists; (2) to integrate the unique local cultural elements and connotations into the renewal of public space through the construction of microspaces, creating a landscape image with local characteristics; (3) to focus on the deep emotional connection between people and places on the basis of meeting people’s basic physiological and psychological needs, developing a sense of place among recreationists; (4) to improve the functional division of parks, enhance the rationality of the layout and meet the needs and preferences of different groups for recreational activities, in order to ensure that recreationists have a pleasant experience in the park and extend their duration of stay; (5) to improve the infrastructure and service management of parks to enhance the willingness of recreationists to revisit and recommend the places.

Second, as for urban management, it is recommended (1) to adopt relevant policies on ticket reduction and exemption, and open the parks to the society for free or at preferential prices to reduce the cost of recreation, thus promoting the positive moderating effect of temporal self-regulation in residents’ travel; (2) to improve the level of public transportation services and optimize the connectivity between various parks and traffic arteries, thus enhancing the convenience and accessibility, increasing the frequency of trips; (3) to pay attention to public mental health, encourage residents to take local trips, introduce preferential measures such as cultural and travel consumption vouchers and advocate restoring individual emotions and energy through recreation in parks; (4) to enhance the openness of urban public spaces, create open green spaces with various characteristics, guide community activities and promote the popularization and normalization of outdoor recreation; (5) to construct a perfect urban nature system and build a network of green spaces to satisfy the requirements of residents to interact with nature and feel more connected to the environment.

### 6.4. Limitations and Future Research

As an exploratory study, the following three aspects can be investigated in future research, due to the limitations of the current research:

First, the data collection was conducted during the comprehensive epidemic prevention and control phase, and the epidemic management requirements and the need to maintain social distance may have an impact on the localization of recreation. The sample should be increased in the future after the end of the epidemic so that the generalizability of the study findings can be improved by comparing data before and after the pandemic.

Second, there is another situation for the localization of recreation: the “residentization” of foreign tourists, which means that foreign tourists behave more like residents in terms of tourism consumption patterns and behavioral preferences. In future research, the differences in behavioral characteristics of localization of recreation and the formation mechanisms of sense of place between foreign tourists and local residents can be compared to enhance the integrity of the theory.

Third, in the design and test of the model, this study creatively drew on the TST framework model proposed by Hall and Fong, which is an extension of the application of TST in the field of recreational behavior, but it failed to comprehensively apply the TST model. In the future, a combination of multiple research methods will be used to verify the applicability of TST in more fields.

Fourth, this study explored the formation mechanisms of sense of place and provided a new opportunity to better understand it, but still failed to fully open the black box of psychological mechanisms of sense of place. Sense of place is a complex theory and future research should add more psychological derivative structures to the existing model in order to explore the inner workings of sense of place to a greater extent.

In a sense, urban recreation space is shaped by globalization and localization, and with the coexistence and interaction of the global and the local, new social and cultural practices emerge, all of which point to the individual’s imagination of “localization” and important formative factors. Therefore, when recreation space becomes uniform due to globalization, the concept of “localization” needs to be reconsidered and reinterpreted. “Localization” should be extended to urban residents’ recreation decisions, based on the needs of residents for local recreation, to explore the spirit of place and the way of self-rest in urban parks, then interpret it in a completely new way and maximize its value, so that recreation can return to the essence of “life”.

## Figures and Tables

**Figure 1 ijerph-20-05160-f001:**
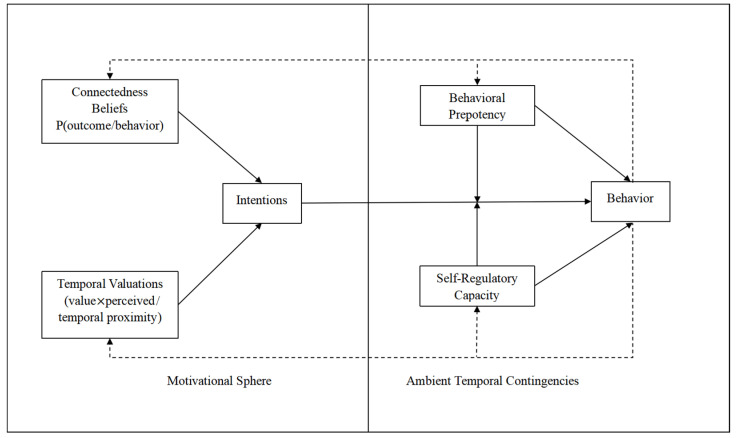
Mechanism of temporal self-regulation theory.

**Figure 2 ijerph-20-05160-f002:**
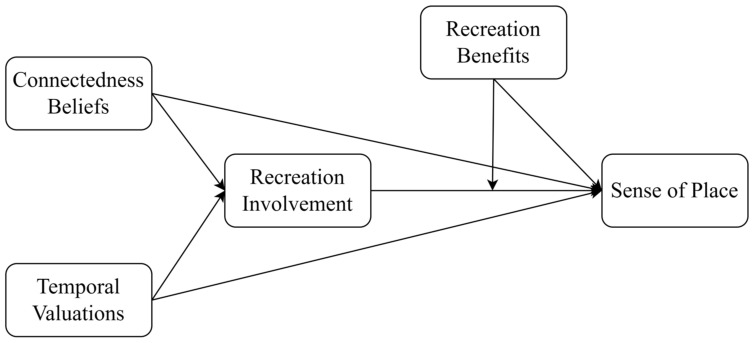
Theoretical model.

**Figure 3 ijerph-20-05160-f003:**
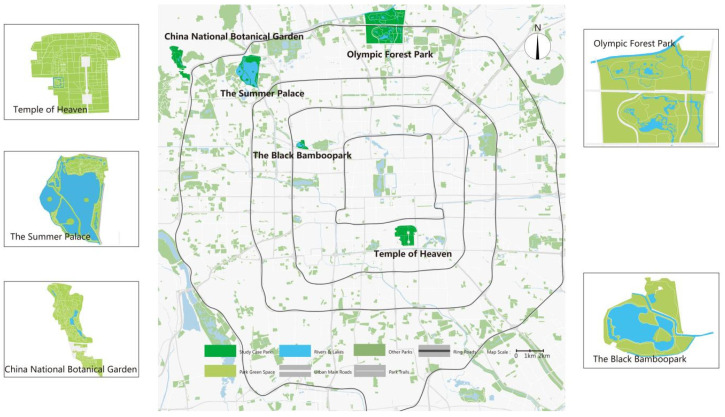
Locations of case parks.

**Figure 4 ijerph-20-05160-f004:**
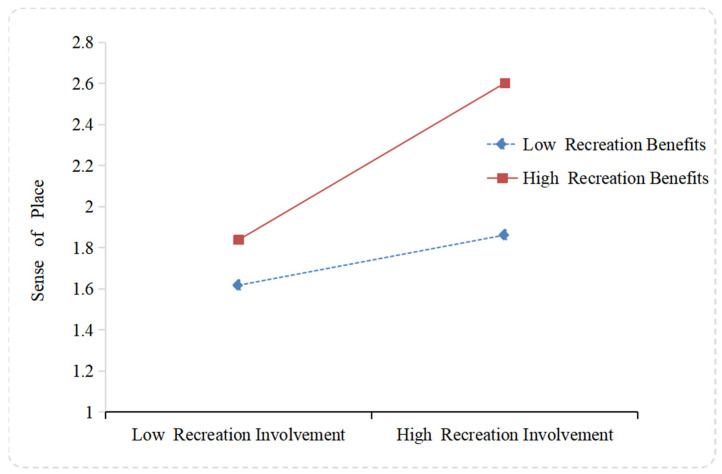
Moderating effect of recreation benefits on the relationship between recreation involvement and sense of place.

**Table 1 ijerph-20-05160-t001:** Socio-demographic characteristics of respondents (N = 545).

Variable	Category	Number	Percentage (%)
Gender	Female	275	50.5
	Male	270	49.5
Age	≤18	109	19.9
	19–30	141	25.9
	31–45	124	22.8
	46–60	120	22
	>60	51	9.4
Physical condition	Good	199	36.5
	Fair	206	37.8
	Poor	140	25.7

**Table 2 ijerph-20-05160-t002:** Recreation behavior characteristics of respondents (N = 545).

Variable	Category	Number	Percentage (%)
Recreation time spent in the park each time	Below 1 h	132	24.2
	1–2 h	192	35.2
	2–3 h	127	23.3
	Over 3 h	94	17.3
Frequency of recreational activities in the park per month	Below 3 times	128	23.5
	3–15 times	224	41.1
	16–30 times	118	21.7
	Over 30 times	75	13.7
Travel mode	Alone	167	30.6
	With family members	154	28.3
	With friends	127	23.3
	Team organization	97	17.8
Travel distance	Neighborhood residents	303	55.6
	Long-distance visitors	242	44.4

**Table 3 ijerph-20-05160-t003:** Results of reliability and convergent validity analysis.

Variable	Item	Standardized Factor Loading Values	Cronbach’s Alpha	C.R.	AVE
Connectedness beliefs	I will feel healthier if I go to the park for recreation.	0.889	0.889	0.888	0.667
	I will feel more comfortable if I go to the park for recreation.	0.805			
	I can strengthen the relationship with family members/friends if I go to the park for recreation.	0.855			
	I will be more hopeful about my future work/life if I go to the park for recreation.	0.706			
Temporal valuations	I will think about the recreational effects of going to the park and make adjustments to my travel plans.	0.831	0.845	0.845	0.646
	I can quickly feel the recreational effects I want by going to the park for recreation.	0.757			
	I am willing to spend more time in the park now for my future health.	0.821			
Recreation involvement	Recreation in the park is a great pleasure for me.	0.854	0.832	0.929	0.687
	I really enjoy the recreation time in the park.	0.855			
	Recreation in the park is important to me.	0.845			
	Recreation in the park reflects a lot about me.	0.791			
	Recreation in the park shows my true self.	0.832			
	I spend most of my recreational life in the park.	0.795			
Recreation benefits	I am willing to share my recreation experiences in the park with others.	0.831	0.904	0.923	0.705
	Going to the park for recreation can alleviate fatigue.	0.845			
	Going to the park for recreation can improve mood.	0.879			
	Going to the park for recreation can improve physical health.	0.821			
	Going to the park can add fun to life and expand interests.	0.821			
Sense of place	I will definitely come to this park for recreation as long as I am available.	0.659	0.884	0.886	0.567
	This park is more suitable for me, although there are other places for recreational activities.	0.774			
	This park allows me to relax myself to the maximum extent.	0.695			
	I really appreciate the environment of this park.	0.807			
	I feel that this park is an essential part of my recreational life.	0.768			
	This park has a special meaning for me.	0.802			

**Table 4 ijerph-20-05160-t004:** Discriminant validity analysis of the scale.

	Connectedness Beliefs	Temporal Valuations	Recreation Involvement	Recreation Benefits	Sense of Place
Connectedness beliefs	0.667				
Temporal valuations	0.357 ***	0.646			
Recreation involvement	0.478 ***	0.404 ***	0.687		
Recreation benefits	0.452 ***	0.466 ***	0.547 ***	0.705	
Sense of place	0.583 ***	0.549 ***	0.645 ***	0.645 ***	0.567
Square root of AVE	0.817	0.804	0.829	0.840	0.753

Note: *** *p* < 0.001. The diagonal is the AVE value.

**Table 5 ijerph-20-05160-t005:** Results of hierarchical regression.

Variable	Sense of Place						Recreation Involvement	
	Model 1	Model 2	Model 3	Model 4	Model 5	Model 6	Model 7	Model 8
Recreation time spent in the park each time	−0.038	−0.032	−0.029	−0.06	−0.049	−0.048	0.038	0.043
Frequency of recreational activities in the park per month	0.024	0.07	0.061	0.038	0.067	0.062	−0.023	0.009
Travel mode	0.023	0.008	−0.011	−0.018	−0.014	−0.026	0.07	0.058
Travel distance	0.014	−0.006	−0.01	0.023	0.007	0.005	−0.016	−0.032
Connectedness beliefs		0.382 ***			0.252 ***			0.338 ***
Temporal valuations		0.366 ***			0.271 ***			0.249 ***
Recreation involvement				0.585 ***	0.384 ***	0.312 ***		
Recreation benefits			0.591 ***			0.296 ***		
Recreation involvement × Recreation benefits						0.237 ***		
R^2^	0.003	0.368	0.349	0.343	0.481	0.491	0.007	0.235
Adj-R^2^	0.005	0.361	0.343	0.337	0.474	0.485	0	0.227
*F*	0.388	52.283	57.72	56.276	71.14	74.063	0.969	27.551

Note: *** *p* < 0.001.

**Table 6 ijerph-20-05160-t006:** Moderating effect of recreation benefits on mediation relationships.

Mediation Path	Recreation Benefits		Effect	BootSE	95% Confidence Interval	
	State	Value			BootLLCI	BootULCI
connectedness beliefs → recreation involvement → sense of place	Low	−1.087	0.0104	0.01	−0.0018	0.036
	Mean	0	0.0434	0.0137	0.0202	0.0723
	High	1.087	0.0972	0.0249	0.0534	0.152
temporal valuations → recreation involvement → sense of place	Low	−1.087	0.0054	0.0126	−0.0158	0.0355
	Mean	0	0.032	0.0146	0.0067	0.0642
	High	1.087	0.0767	0.0224	0.0372	0.1246

Note: 95% CI for conditional direct and indirect effect using bootstrap (bias corrected).

## Data Availability

The data used to support the findings of this study are available from the author upon request.

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
