# Peer review of "A Study on the Localization of Urban Residents’ Recreation: A Moderated Mediation Model Based on Temporal Self-Regulation Theory"

_ijerph, 2023, doi:10.3390/ijerph20065160_

Round 1
Reviewer 1 Report
Dear authors,
The article is a good work and its conclusions would be interesting for further research. However, I would like to advise you on some issues to improve the quality of the manuscript.
Question 1:
The introduction is clear and all objectives are well stated. At the same time, it would be beneficial for the reader if the authors included a comprehensive literature review beforehand in order to have a complete picture of the state of the art.
Question 2:
The theoretical model presented seems coherent, although it would be interesting to determine more and better the practicality of TST in other studies.
I would suggest using PLS software instead of SPSS and compare results.
I would explain a bit more in detail what are the advantages/disadvantages of your proposal and the robustness of the model.
Question 3:
The paper could be improved in terms of examples of application to other situations. It would provide some real applications and limitations of other similar models.
Question 4:
Could new moderating variables be raised in the model such as age or gender?
It would be interesting to compare pre- and post-pandemic data.
Highlight further the limitations of your proposal.
Question 5:
You could have discussed further improvements/future work. Please elaborate on this point.
Author Response
Point 1: The introduction is clear and all objectives are well stated. At the same time, it would be beneficial for the reader if the authors included a comprehensive literature review beforehand in order to have a complete picture of the state of the art.
Response 1: We thank the reviewer for pointing this out. we have made a complete revision of the literature review,The specific details are not listed here,Please check the resubmitted manuscript.
Point 2: The theoretical model presented seems coherent, although it would be interesting to determine more and better the practicality of TST in other studies.I would suggest using PLS software instead of SPSS and compare results.I would explain a bit more in detail what are the advantages/disadvantages of your proposal and the robustness of the model.
Response 2:Thanks to the reviewers' advice, We couldn’t agree more,we decided to use PLS software in the next study.
Point 3: The paper could be improved in terms of examples of application to other situations. It would provide some real applications and limitations of other similar models.
Response 3:Thank you for your comment. We have revised. The specific details are not listed here. Please check the resubmitted manuscript. Thank you for providing hints and comments for our modification.
Point 4: Could new moderating variables be raised in the model such as age or gender?It would be interesting to compare pre- and post-pandemic data.
Highlight further the limitations of your proposal.
Response 4:We thank the reviewer for pointing this out. We have revised. In Planning and management implications and Limitations and future research section, we have added more details.We consider adding moderating variables(age or gender) in the next paper.
Point 5: You could have discussed further improvements/future work. Please elaborate on this point.
Response 5:Thank you for your comment. We have revised. The specific details are not listed here. Please check the limitations and future research section.

Reviewer 2 Report
Overall I appreciate the quality of the paper. There is a large body of literature used and the results are very well linked to the literature review.
Still - I would shorten some bodies of the article, as I proposed on the file attached to this sections.

Author Response
Point 1: Line 41 - Residents' recreation behaviors are based on local life. (not necessary – you repeat it in the following sentence)
Line 60-62 – You need citation for this phrase
Line 64-86 – It needs to be shortened – aiming at introducing the main goal and objectives of the research – moreover – there are sentences that need to be related to other research papers such as
- Residents' sense of 71 place is a complicated and dynamic process that results from the long-term interaction 72 between residents and recreation areas (citation needed)
Literature review – Sense of place- while there is a academic debate related to the sense of place and attractiveness of the recreational facilities – the current paper aims at “investigate the formation mechanism of the sense of place based on recreation involvement and recreation benefits from the perspective of localization of recreation.” Therefore I suggest focusing on the literature and academic paper which
is strictly related to the aim of the paper. Please rephrase 89-105 lines according to the initial aim of the paper.
Response 1:We thank the reviewer for pointing this out,we agree and have updated. Line 41:We have removed the repeated sentence
Line 60-62 :We have updated the references
Line 64-86:We thank the reviewer for pointing this out. we have made a complete revision of the Introduction,The specific details are not listed here,Please check the resubmitted manuscript.
Point 2: Literature review 2.2 Temporal self-regulation theory – I suggest shortening the entire subchapter as it is not the intention to describe the TST but rather to relate your research with the theory.
Response 2:Thank you for your comment, We have revised.Literature review 2.2 has been revised across the research to be more relevant to this study.
Point 3: For Chapter 3 Research Hypothesis. I find them very well structured and well connected to the aim of the paper. However, much of the description for each hypothesis is unnecessary as this will either fit better at chapter 2 (Literature review) or can be reduced for having a better focus on the initial hypothesis. I suggest having one chapter, without subchapters and reducing the amount of phrases they intend to be partially part of literature review and partially a base for creating the hypothesis.
Response 3:We thank the reviewer for pointing this out.Due to the length of the article, Chapter 3 could not be merged into Chapter 2, but we reduced some content to make it more concise.
Point 4: For chapter 4.2 – Case parks – Even though in the first part of the article there is a large amount of literature cited related to the localization and the characteristics of the parks, the description of the study cases is rather scarcely. I would suggest adding two phrases which are related to the position of the parks related to the surroundings. Which area do they belong? Residential, mixt, administration? Who could be the possible visitors, according to the proximity?
Response 4:This observation is correct, and we have modified, we added:
Specifically, comprehensive parks refer to parks with complete functions, well-equipped facilities, and rich content, which can satisfy the diverse needs of different groups of visitors. Community parks refer to parks with necessary supporting facilities and activity areas, mainly serving the residents within a certain residential area for daily leisure activities. Historical parks refer to parks with outstanding historical and cultural value, which have had an impact on the urban transformation or cultural and artistic development of the city. Ecological parks refer to parks with natural environments characterized by original ecology or low human interference, which focus on meeting visitors' needs to get close to nature, including forest parks, suburban parks, wetland parks, and so on. Cultural theme parks refer to parks with special themes or cultures as their core content, including theme parks, botanical gardens, zoos, and amusement parks.
Point 5: For Chapter 5 – Results – Line 413 – You already have this statement in line 382, no need to add second time.
Line 508-510 – You need a citation for this statement.
Line 567-576 – While you have a very valid question -The first phrase (567-570) is not related to the following ideas. Therefore, I suggest rephrasing and bringing a strong argument for the question.
Response 5:We thank the reviewer for pointing this out.
Line 413:The first occurrence of this sentence is about the data analysis of the pre-survey, and the second occurrence is about the data analysis of the formal survey. Based on the academic paradigm, we did not delete this sentence in the second occurrence
Line 508-510:We thank the reviewer for pointing this out,We have updated the references.
Line 567-576:We thank the reviewer for pointing this out. We couldn’t agree more, We rephrased for the phrase,
The process of localization takes place in the context of the extensive connection between people and places, whose abundant habitual practices constitute substantial internal diversity, while the habitual practices based on places are intimately embedded in a broad spatial structure of social relations.
Point 6: Line 592-621 – While there is obvious practical implications – these are rather very general conclusions which are not directly linked to the results of the present research. I would either shorten this subchapter or propose a better connected practical implications
Response 6:We thank the reviewer for pointing this out. we have made a complete revision of 6.3,The specific details are not listed here,Please check the resubmitted manuscript.

Reviewer 3 Report
The authors introduce a moderated mediation model to study the localization of urban residents' recreation. I want to provide some comments for the authors.
1. There is lacking novelty and fun, and moreover, urgency. Does the localization only show in urban residents' recreation? Do urban residents' recreation in Beijing the same as urban residents in other cities worldwide?
2. Why TST is the most important and needed theory? There are many behavioral theories you have to discuss. The authors indicated that "most studies have found that health-related behaviors are not completely under intentional control[20]", the cited reference is too old to persuade us.
3. There is some miss information between hypothesis inference and research framework. For example, paths H1a and H1b didn't show in the framework.
4. What is the sampling method? How is the sample representativeness?
5. The conclusion should compare to previous studies.
6. Please provide social implication and economical implication.
7. Many errors with inconsistent details appear in the list of references.
8. I suggest the authors try to ask native English speakers to polish this manuscript.
Author Response
Point 1: There is lacking novelty and fun, and moreover, urgency. Does the localization only show in urban residents' recreation? Do urban residents' recreation in Beijing the same as urban residents in other cities worldwide?
Response 1:We thank the reviewer for pointing this out. We couldn’t agree more, the localization of recreation is not just the exclusive discourse of urban residents' recreation.In future research, the differences in behavioral characteristics of localization of recreation and the formation mechanisms of sense of place between foreign tourists and local residents can be compared to enhance the integrity of the theory.
We chose to collect data in urban parks of Beijing for two reasons:(1)Beijing has many urban parks which share obvious characteristics of large scale, diversity and balanced development, providing both tourism services for foreigners and leisure services for local residents. (2)The five case parks attract crowds of recreationists and serve a wide range of people, forming high-quality places for recreation with certain social influence.Therefore, they are very representative and typical in reflecting recreation behaviors.
Point 2: Why TST is the most important and needed theory? There are many behavioral theories you have to discuss. The authors indicated that "most studies have found that health-related behaviors are not completely under intentional control[20]", the cited reference is too old to persuade us.
Response 2:We thank the reviewer for pointing this out. we agree and updated the references.
Hagger M S, Hardcastle S J, Chater A, et al. Autonomous and controlled motivational regulations for multiple health-related behaviors: between-and within-participants analyses. Health Psychology and Behavioral Medicine, 2014, 2, 565-601.
Point 3: There is some miss information between hypothesis inference and research framework. For example, paths H1a and H1b didn't show in the framework.
Response 3:We thank the reviewer for pointing this out. We have modified Figure 2.
Point 4: What is the sampling method? How is the sample representativeness?
Response 4:Recreationists were chosen using systematic random sampling in areas with numerous recreationists,very five recreationists passing through the areas were approached. Before conducting the questionnaire research, an optional question "Are you a local resident" was set to filter out non-target objects. After the recreationists had accepted the invitation, one research assistant informed them of the purpose of the survey, the confidentiality of information, and the meanings of some incomprehensible concepts.
Point 5: The conclusion should compare to previous studies.
Response 5:We thank the reviewer for pointing this out. We couldn’t agree more, We have revised. The specific details are not listed here. Please check the resubmitted manuscript.
Point 6: Please provide social implication and economical implication.
Response 6:We agree and have updated,Please check 6.3 Planning and management implications
Point 7: Many errors with inconsistent details appear in the list of references.
Response 7:We thank the reviewer for pointing this out. we agree and updated the references.
Point 8: suggest the authors try to ask native English speakers to polish this manuscript.
Response 8:Thank you for your comment. we have made modifications to the English review, The specific details are not listed here. Please check the manuscript. Thank you very much for your effort on this article.

Reviewer 4 Report
The paper constructed a moderated mediation model based on temporal self-regulation theory, by developing a case study on five representative urban parks in Beijing.
The data collected through 545 valid questionnaires were used to discuss the behavioral characteristics of localized recreation and the formation mechanism of sense of place among urban residents in Beijing.
The paper addresses a relevant topic, up-to-date and adequately supported by the literature. It is well organised and accompanies the reader's flow of reasoning.
However, a few improvements can be included as follows:
Introduction:
Please consider to better discuss the “recreation” activities and the implication for urban renewal including references to urban planning literature.
Research hypotheses:
The sentence in line 186-190 is unclear
§ 6.3. The title could be modified in “Plannibng and management implications”
Please check some mistyping such as in line 623, 630
Author Response
Point 1: Introduction:Please consider to better discuss the “recreation” activities and the implication for urban renewal including references to urban planning literature.
Response 1:Thank you for your comment, we have made a complete revision of the introduction,The specific details are not listed here,Please check the resubmitted manuscript.
Point 2: Research hypotheses:The sentence in line 186-190 is unclear
Response 2:We thank the reviewer for pointing this out. we agree and updated the sentence.
By specifying behavioral self-regulation as influenced by both motivational and momentary factors, TST not only offers the potential to improve the prediction of behavior but also to identify pre-and post-intentional factors influencing successful behavioral self-regulation which could be targeted by interventions. In light of previous research and the tenets of TST, this study aims to explore the constructs of TST as predictors of sense of place. Five variables are integrated into the TST framework, namely, recreation involvement into intention valuation, recreation benefits into perception evaluations where behavioral prepotency and self-regulatory capacity work together, and sense of place into behavioral intention.
Point 3:The title could be modified in “Planning and management implications”
Please check some mistyping such as in line 623, 630
Response 3:We thank the reviewer for pointing this out. We couldn’t agree more, We modified the title
